# A Six-year circum-Antarctic icebergs dataset (2018-2023)

Zilong Chen<sup>1,2,\*</sup>, Xuying Liu<sup>3,\*</sup>, Zhenfu Guan<sup>1</sup>, Teng Li<sup>1,2</sup>, Xiao Cheng<sup>1,2</sup>, Tian Li<sup>4</sup>, Yan Liu<sup>5</sup>, Qi Liang<sup>1,2</sup>, Lei Zheng<sup>1,2</sup>, and Jiping Liu<sup>6</sup>

<sup>1</sup>School of Geospatial Engineering and Science, Sun Yat-sen University, Southern Marine Science and Engineering Guangdong Laboratory (Zhuhai), Zhuhai 51908, China
<sup>2</sup>Key Laboratory of Comprehensive Observation of Polar Environment (Sun Yat-sen University), Ministry of Education, Zhuhai 519082, China
<sup>3</sup>Institute of Artificial Intelligence, Shaoxing University, Shaoxing 312000, China
<sup>4</sup>Bristol Glaciology Centre, School of Geographical Sciences, University of Bristol, Bristol BS8 1SS, UK
<sup>5</sup>State Key Laboratory of Remote Sensing and Digital Earth, College of Global Change and Earth System Science, Beijing Normal University, Beijing 100875, China
<sup>6</sup>School of Atmospheric Sciences, Sun Yat-sen University, Zhuhai 51908, China

Correspondence: Teng Li (liteng28@mail.sysu.edu.cn) and Xiao Cheng (chengxiao9@mail.sysu.edu.cn)

**Abstract.** The distribution of Antarctic icebergs is crucial for understanding their impact on the Southern Ocean's atmosphere and physical environment, as well as their role in global climate change. Recent advancements in iceberg databases, based on remote sensing imagery and altimetry data, have led to products like the BYU/NIC iceberg database, the Altiberg database, and high-resolution SAR-based iceberg distribution data. However, no unified database exists that integrates various iceberg scales

- and covers the entire Southern Ocean. Our research presents a comprehensive circum-Antarctic iceberg dataset, developed using Sentinel-1 SAR imagery from the Google Earth Engine (GEE) platform, covering the Southern Ocean south of 55°S. A semi-automated classification method that integrates incremental random forest classification with manual correction was applied to extract icebergs larger than 0.04 km<sup>2</sup>, resulting in a dataset for each October from 2018 to 2023. The resulting dataset not only records the geographic coordinates and geometric attributes (area, perimeter, long axis, and short axis) of
- the icebergs but also provides uncertainty estimates for area and mass. The dataset reveals significant interannual variability in iceberg number and total area-the number of icebergs increased from 33,823 in 2018 to approximately 51,332 in 2021, corresponding to major ice shelf calving events (e.g., the A68a iceberg in the Weddell Sea), followed by a decline in 2022. The annual average total iceberg area is 44,518  $\pm$  4800 km<sup>2</sup>, and the average mass is 8,779  $\pm$  3,029 Gt. Validation using test set samples and a rolling cross-validation of interannual variability shows that the integrates incremental random forest
- classification achieves accuracy, recall, and F1 scores exceeding 0.90, and after manual correction, all performance metrics should be even better. Comparisons with existing iceberg products (including the BYU/NIC iceberg database and the Altiberg database) indicate a high consistency in spatial distribution, while our dataset demonstrates clear advantages in terms of spatial coverage, iceberg detection scale, and identification capabilities in regions with dense sea ice. This dataset serves as a novel data resource for investigating the impact of Antarctic icebergs on the Southern Ocean, the mass balance of ice sheets, the
- mechanisms underlying ice shelf collapse, and the response mechanisms of iceberg disintegration to climate change.

# 1 Introduction

Icebergs are large freshwater ice masses that break off from the edges of ice sheets, ice shelves, or glaciers and enter the ocean. They are a critical component in the global climate system (Benn and Åström, 2018). Approximately half of the mass loss from the Antarctic ice sheet is discharged into the Southern Ocean through iceberg calving (Depoorter et al., 2013; Rignot et al., 2013; Liu et al., 2015). Annually, the dissolution of over 100,000 icebergs into the ocean is estimated to introduce a volume of freshwater that, according to certain calculations, exceeds the global annual freshwater consumption (Qadir et al.,

2022; Orheim et al., 2023). This resultant freshwater influx plays a critical role in influencing the thermohaline characteristics, heat content, and freshwater balance within the impacted regions of the Southern Ocean (Gladstone et al., 2001; Hammond and Jones, 2016). On the bottom, grounding icebergs can interact with ocean floor and leave scours as a kind of geological

- record (Dowdeswell and Bamber, 2007; Li et al., 2018; Liu et al., 2021). Additionally, the nutrients carried by icebergs can influence the spatial distribution of primary productivity (Duprat et al., 2016), promoting the development of local ecosystems (Smith et al., 2007; Wu and Hou, 2017; Lin et al., 2024). Furthermore, icebergs pose a potential threat to maritime activities (Bigg et al., 2018), as human activity in the Antarctic region increases, accurate monitoring of iceberg distribution, size, and trajectory prediction has become critical (Evans et al., 2023)
- The current databases on the distribution of Antarctic icebergs, as shown in Table 1, are primarily categorized into four types: (1) Ship-based observations, such as the SCAR International Iceberg Database (Orheim et al., 2023), compiled and published by the Norwegian Polar Institute (NPI) and the Scientific Committee on Antarctic Research (SCAR), which records 323,520 icebergs and serves as an important historical dataset. However, it is only confined to shipping lanes, not fully representing the Antarctic iceberg's spatial distribution and its interannual changes; (2) Low-resolution satellite imagery-based databases,
- with the National Ice Center (NIC) and Brigham Young University (BYU) Antarctic Iceberg Database as a notable example (Long et al., 2002; Stuart and Long, 2011a, b). Budge and Long (2018) consolidated these databases to offer iceberg location, length, and area data, but they are restricted to larger icebergs (length>5km) due to the limitations of low-resolution imagery; (3) Satellite radar altimetry-based databases, like the Altiberg database from the French Research Institute for Exploitation of the Sea (Tournadre et al., 2012, 2015, 2016, 2024). This database is effective at detecting icebergs in open waters, but in
- complex scene, such as areas with dense ice or high iceberg concentrations, it becomes challenging to extract accurate iceberg information from the altimetric waveforms; (4) High-resolution SAR data-derived products. Wesche and Dierking (2015) extracted icebergs larger than 0.3 km<sup>2</sup> in the Antarctic coastal region using Radarsat-1 circum-Antarctic mosaic images. Barbat applied a random forest algorithm to Radarsat circum-Antarctic mosaic images from 1997, 2000, and 2008 to obtain iceberg distributions for the corresponding years (Barbat et al., 2019a); (5) circum-Antarctic iceberg calving dataset. This dataset was
- derived from continuous optical (MODIS and Landsat-8) and radar (Envisat ASAR and Sentinel-1) satellite observations and was released by Qi et al. (2021). The product provides detailed information on each calving event, including time, area, size, thickness, etc., but it only focused on the transient icebergs just calved from ice shelves therefore lacking the spatial distribution across the open ocean. All above data products primarily cover the Antarctic coastal region, and the published datasets are not

Table 1. Overview of Antarctic Iceberg Datasets.

| Iceberg dataset         | Time scale          | Iceberg size range       | Satellite data(sensors)                        |
|-------------------------|---------------------|--------------------------|------------------------------------------------|
| The SCAR Interna-       | 1982-2010           | >10m                     | -                                              |
| tional Iceberg Database |                     |                          |                                                |
| USNIC Antarctic Ice-    | 1978-Present        | >18 km                   | SAR, visible, and infrared remotely sensed im- |
| berg Data               |                     |                          | agery                                          |
| BYU Antarctic Iceberg   | 1978 & 1992-Present | >5 km                    | SASS, ERS-1/2, NSCAT, QuikSCAT, ASCAT,         |
| Tracking Database       |                     |                          | OSCAT, SeaWinds, NIC (multiple sensors)        |
| Altiberg                | 1992-2023           | Determined by the res-   | ERS1/2, Topex, Poseidon, Jason1/2/3, Envisat,  |
|                         |                     | olution of the satellite | Cryosat, Cryosat SAR, Cryosat SARIN, AL-       |
|                         |                     | altimeter                | TIKA, HY-2A/B/C, Sentinel-3(A&B) PLRM,         |
|                         |                     |                          | Sentinel-3(A&B) SAR, Geosat                    |
| Qi et al., 2021         | 2005-2020           | >1 km                    | Envisat ASAR, Sentinel-1 SAR, MODIS,           |
|                         |                     |                          | Landsat 8 OLI                                  |

real-time monitoring results, but rather used for historical scientific research. In summary, there is currently no comprehensive

iceberg database covering multiple scales and the entire Southern Ocean has been established to date.

High-precision, large-scale, and long-term continuous remote sensing observations of circum-Antarctic iceberg distribution not only characterize the spatiotemporal patterns of iceberg occurrence but also provide critical data for elucidating the mechanisms of iceberg formation and evolution, ice-shelf dynamics, and their complex interactions with climate change. In this study, we leveraged the Google Earth Engine (GEE) platform to acquire Sentinel-1 SAR mosaic imagery and applied an

- incremental random forest classification combined with manual correction to identify Antarctic icebergs larger than 0.04 km<sup>2</sup>, extracting each iceberg's outline, location, area, mass, and associated uncertainty. Based on these results, we constructed a circum-Antarctic iceberg distribution dataset covering each October from 2018 to 2023 and conducted a comprehensive analysis of the spatiotemporal characteristics of iceberg distribution over this six-year period. To ensure the reliability of the dataset, we performed an internal accuracy validation of the classifier and conducted external validation by comparing our results with existing iceberg databases and data products.
- bb existing reeberg databases and data prod

# 2 Data

To identify circum-Antarctic icebergs, we utilized the European Space Agency (ESA) Sentinel-1 C-band SAR Ground Range Detected (GRD) data. Given the extensive coverage of the data, we chose the Extra Wide (EW) swath mode, which provides a spatial resolution of 40 m. The Sentinel-1 data offers various band combinations based on different polarization modes (e.g.,

VV, HH, VV + VH, and HH + HV), with HH polarization being the primary mode available in polar regions (Koo et al., 2023;
 Ferdous et al., 2018). Therefore, only HH polarization band images were used for analysis.