# Peer review of "A Six-year circum-Antarctic icebergs dataset (2018-2023)"

_Earth System Science Data, 2025_

## Author Comment (AC1)

This paper presents a circum-Antarctic iceberg database using Sentinel-1 SAR images in the Google Earth Engine platform. Their image segmentation and random forest classifier seem to work successfully in capturing the spatiotemporal distributions of icebergs, including their number and sizes, across the Southern Ocean. However, the authors need to provide more details about their iceberg detection model. While the authors mentioned that they used an ensemble random forest classifier with four different RF classifiers, based on different input features, they did not provide any details about this ensemble result (i.e., weights to each classifier, importance of statistical features, histogram features, and texture features). I encourage the authors to provide the details of their ensemble process to support the robustness of their method.

Thank you for your comments and suggestions. In the original manuscript's Method, we did not describe the model's specific parameters in detail. In the revised manuscript, we have added more detailed information on the model integration methods (L158-L172). Below are our point-by-point responses to your comments:

L146-147: How are these three subsets divided? Randomly or by any other criteria?

The sample set is randomly divided into three subsets, and we have added the relevant explanation in the revised manuscript (L155).

L210: Maybe it would be better to use 40 m, instead of 0.04 km, as already used throughout the manuscript (L69 and L216).

Thank you for your suggestion. To maintain consistency with the surrounding text, it is better to use "40 m".

L241: "Based on this analysis, we selected an average thickness of 232 m for the icebergs" -> It is not clear how this value of 232 m is derived.

This study used 19,945 iceberg freeboard measurements from the Altiberg v3.2 dataset recorded by the CryoSat SARIn mode during 2018–2021 (the latest year available in this version), yielding an average freeboard of approximately 40 m, which was adopted as the representative freeboard for all icebergs in this study. Following previous research, we set the seawater density to 1025 kg/m³ and the iceberg density to 850 kg/m³. According to Archimedes' principle, the relationship between iceberg freeboard height h and total thickness H is:

$$H = \frac{\rho_w}{\rho_w - \rho_i} h$$

Substituting the above parameters into the equation, the average total thickness of icebergs is calculated to be 232 m.

In the revised version, instead of assigning a uniform fixed thickness of 232 m to all Antarctic icebergs, we assign thickness values according to iceberg area (L205-213), based on the Volume/Area scaling parameterization of Iceberg Classes Model in Stern et al. (2016), thereby making the thickness attribute of individual icebergs more physically meaningful.

L256-259: Then, does it mean that 2018 data was included in training for all iterations but not tested at all, and 2023 data was never used for training? If so, I don't think this is a fair training strategy because the model could be biased to 2018 data. Would it be better to conduct 6-fold cross-validation (or so-called Leave-One-Out cross-validation), for example, 2018 data as test data and the remaining years as training data for iteration 1, 2019 data as test data and the remaining years as training data for iteration 2, and so forth? The authors mentioned that they used

this strategy to "adapt to the time-series nature of the data while minimizing the risks of overfitting" (L256), but I'm not sure how the current strategy can achieve this.

Thank you for the reviewer's thorough comments. We fully agree that leave-one-year-out cross-validation ensures a fair assessment of the model's performance on each year's Sentinel-1 imagery. Accordingly, in the revised manuscript we have adopted a six-fold, leave-one-year-out cross-validation scheme, using approximately 400 manually annotated superpixel samples from each year as the test set and the remaining years' data for training, thereby ensuring that each year both tests and contributes to the training. We have replaced the original each year evaluation and rolling window validation results in the main text (L262-265) and in Table 3. The new table presents the Accuracy, Precision, Recall, and F1 score of the ensemble incremental random forest classifier with optimal parameters for each year from 2018 to 2023, as well as their averages, to more comprehensively demonstrate the model's accuracy and cross-year generalization ability in circumpolar Antarctic iceberg detection.

Tables 3 and 4: The authors conducted performance evaluations twice: (i) evaluation for each year (Table 3) and (ii) evaluation with rolling window validation (Table 4). I'm not sure that these two different evaluations are really necessary. To evaluate the model performance, I believe cross-validation in Table 4 is enough.

Thank you for the reviewer's suggestion. In the original manuscript, we employed both annual evaluation and rolling-window validation: the former to demonstrate classification performance on each year's Sentinel-1 imagery, and the latter to illustrate the model's robustness as historical data accumulate. To streamline the manuscript, we have

adopted the reviewer's recommendation, revised the evaluation strategy to leave-one-year-out cross-validation, and retained only these results in the main text.

L263-264: So, what model is finally used for building the iceberg database? The database is built each year separately based on the random forest model in Table 3, or does the entire database use a single model trained from the final iteration in Table 4?

We ultimately adopted the random forest models from Table 3, using distinct parameter settings for each year, and constructed the iceberg database separately for each year.

Section 4.1: The authors should have provided a detailed performance of their "ensemble" RF model. In L150-154, the authors mentioned that they used four RF classifiers and assigned weights to these classifiers, but the manuscript lacks details about this process. It is necessary to specify the performance of these four classifiers and how the authors select the weights between these models.

Thank you for the referee's valuable suggestion. Indeed, the original manuscript lacked a detailed description of the model's specific parameters, so we have supplemented this information in the revised manuscript (L158-172). Below, we take October 2018 as an example to illustrate how we determined the parameters for our ensemble random forest classifiers.

Based on the Sentinel-1 SAR imagery, we applied the SLIC algorithm to generate superpixels and then manually selected approximately 2,000 superpixel samples per year, with roughly half representing icebergs and the remainder non-icebergs. The sample set was then randomly divided into three subsets: an initial training set, a

validation set, and a test set, in a 6:2:2 ratio. The training set was used to train the RF classifier, the validation set was used to evaluate the model's performance and optimize parameters, and the test set was used for final evaluation of the model's generalization ability and reliability.

Taking October 2018 as an example, we detailed how we determined the parameters for our ensemble of random forest classifiers and performed an incremental training procedure within each 5°×5° grid cell. We constructed four independent random forest models: RF1 trained on statistical features, RF2 on histogram features, RF3 on texture features, and RF4 on all combined features. By analyzing out-of-bag error (OOB) curves under various hyperparameter settings, we identified the configurations that converged stably with minimum OOB: 200 trees/3 features for RF1, 100 trees/5 features for RF2, 250 trees/7 features for RF3, and 150 trees/3 features for RF4 (Fig. S1). Each model was then evaluated on the validation set to compute accuracy, precision, recall, and F1 score (Table S1), these four metrics were normalized to generate candidate weight schemes reflecting different perspectives on sub-model importance (Table S2).

For the ensemble, we multiplied each model's iceberg probability by its corresponding weight and summed the results to obtain a combined discriminant score for each superpixel. We scanned decision thresholds from 0 to 1 in steps of 0.01 on the validation set, plotting precision–recall and ROC curves for each weight scheme. The scheme that maximized the sum of P–R Area Under the Curve (AUC) and ROC AUC was selected as optimal, yielding weights of 0.218, 0.271, 0.246 and 0.265 for RF1–RF4, respectively. Finally, we searched for the threshold that maximized the F1 score on the validation set and set 0.783 as the final decision threshold for iceberg detection.The same procedure was applied to the remaining years to obtain the optimal parameter configurations for each respective year.

[Figure]

Figure S1. Out-of-bag error and parameter importance of random forest classifiers based on different feature sets.

Table S1. Performance metrics of random forest classifiers based on different feature sets

| classifier | ACC | Precision | Recall | F1 |
|---|---|---|---|---|
| RF1 | 0.9207 | 0.9505 | 0.8872 | 0.9178 |
| RF2 | 0.9872 | 0.9847 | 0.9897 | 0.9872 |
| RF3 | 0.9488 | 0.9534 | 0.9436 | 0.9485 |
| RF4 | 0.9872 | 0.9948 | 0.9795 | 0.9871 |

Table S2 Normalized weights of random forest classifiers derived from different evaluation metrics

| weight | RF1 | RF2 | RF3 | RF4 |
|---|---|---|---|---|
| ACC | 0.2293 | 0.2636 | 0.2435 | 0.2636 |
| Precision | 0.2396 | 0.2571 | 0.2410 | 0.2623 |
| Recall | 0.2176 | 0.2709 | 0.2462 | 0.2653 |
| F1 | 0.2282 | 0.2641 | 0.2437 | 0.2640 |

L300: "several tens of kilometers: This is too ambiguous. Please provide specific numbers.

We thank the referee for their careful correction. The specific values here are 44.08 km and 32.28 km, and we have amended this in the revised manuscript (L304).

L301-303: I would like to ask the authors to provide more details about why the BYU/NIC database cannot capture so many > 5 km icebergs. Does it intentionally skip relatively small icebergs (near 5 km size), or does its iceberg detection algorithm, by itself, have limitations in capturing near-5-km icebergs? What about much larger icebergs, for

example, > 10 km?

Taking 2021 as an example, we downloaded iceberg trajectory data (Statistical Database [v7.1]) from the official Brigham Young University website (https://www.scp.byu.edu/data/iceberg/default.html), comprising 192 records representing the observed iceberg trajectories during the same period. we extracted the entries in the csv file whose "date" field corresponds to October 2021, obtaining 53 records for comparison with our study's iceberg spatial distribution data. to ensure comparability between the two datasets, we also selected from our database all icebergs with a major axis exceeding 5 km (292 in total, of which 88 exceed 10 km in major axis). during the comparison, we followed the iceberg position and shape reports published by the u.s. national ice center and applied a one-to-one matching approach to rigorously verify each iceberg's spatial location and shape characteristics.

The results show that all 50 icebergs recorded by BYU/NIC for October were matched in our database (black boxes in figure 2), while three icebergs (C36, B46, and UK324) were not detected (red boxes). further analysis indicates that C36 and B46 were located in the sentinel-1 SAR EW scan-mode blind zone, which remained uncovered even after image mosaicking. for UK324, no iceberg with a major axis exceeding 5 km was found in the corresponding mosaic or original single sentinel-1 images, suggesting potential positioning or identification errors in the BYU/NIC record. to improve the completeness and accuracy of our dataset, we supplemented the EW blind zone with sentinel-1 IW-mode data (Fig. S2 C63).

Fig. S3 presents the spatial distribution of icebergs with a major axis larger than 5 km detected in our study but not recorded in the BYU/NIC database, overlaid on the sentinel-1 mosaic image used in our analysis. it is evident that no duplicate counts or incorrectly merged icebergs

occurred. figure 4 further illustrates the distribution characteristics of these icebergs in terms of area and major axis length: the number of icebergs decreases markedly as area and major axis increase, with most icebergs having an area between 0 and 20 km² and a major axis within the 5–9 km range. spatially, these icebergs undetected by BYU/NIC are mainly located in front of ice shelves and are typically accompanied by sea ice cover.

According to Budge and Long (2018), the BYU/NIC database has several limitations that can result in the omission of even large icebergs with major axes exceeding 5 km. First, the database primarily relies on passive microwave and scatterometer data for tracking. These sensors have relatively low spatial resolutions, typically on the order of several to tens of kilometers, so in areas with dense sea ice cover or high iceberg concentrations, the signal from an individual iceberg can easily blend with surrounding targets, leading to missed or false detections. Second, both the automatic and manual identification processes in the BYU/NIC database can be affected by cloud cover, wind waves, and other anomalous electromagnetic scattering conditions. In particular, in the complex environments in front of ice shelves or along coastlines, the signals from large icebergs may be obscured by sea ice, making them difficult to distinguish. In addition, due to temporal gaps in observational coverage, to maintain consistent measurement intervals in the consolidated database, researchers perform piecewise cubic interpolation of iceberg positions between consecutive observations, while no

interpolation is conducted for observation gaps longer than two weeks. Although this approach can partially fill short-term data gaps, it may lead to inaccurate position estimates or even omissions from the records for large icebergs that drift rapidly or disintegrate within a short period of time.

[Figure]

● BYU/NIC    ☐ our research

Figure S2. spatial matching results of icebergs between the BYU/NIC database (red dots) and our dataset (yellow polygons): black boxes denote

successfully matched icebergs; red boxes denote unmatched icebergs.

[Figure]

☐ Icebergs(> 5 km) missing from BYU/NIC database
☐ Icebergs detected by BYU/NIC database

Figure S3. Examples of icebergs (>5 km) detected (blue) and missed (red) by the BYU/NIC dataset, with Sentinel-1 mosaics as background.

[Figure]

Figure S4. Histogram distribution of the area and major axis length of icebergs (>5 km) missed by the BYU/NIC dataset.

L339-349: I wonder if the total number of icebergs here and in Table 5 is the "true" number of icebergs. That is, if an iceberg is detected in two different Sentinel-1 scenes, how is this iceberg counted? This iceberg could be counted in duplicate, as the methods proposed in this study can only "detect" icebergs but cannot "track" identical icebergs. This could not be so significant because the authors used mosaiced data, but there is a possibility that the same icebergs are detected in duplicate (or some icebergs are missed) due to their drift even over a short period. It would be worthwhile to mention this issue and include any relevant discussion about it.

We greatly appreciate the referee's valuable suggestions! In this study, we believe that the total iceberg counts listed in lines 339–349 of the original manuscript and in Table 5 already reflect the real situation as accurately as possible. First, during the image-acquisition stage we sorted all Sentinel-1 HH-band images within each tile in ascending order of acquisition time and then mosaicked them sequentially using the mosaic()

function in Google Earth Engine. Later-acquired images overwrite valid pixels in earlier images, filling voids at the beginning of the month and producing a synthetic layer that is both spatially continuous and representative of the month's most recent observations. Because of this time-ordered mosaicking approach, the intervals between dates of the images composited into any single tile are generally small.

Taking 2021 as an example, we analyzed all 360 Antarctic tiles (280 after excluding no-data tiles) in terms of the number of distinct acquisition dates and the span between the earliest and latest dates (Figure S5). The results show that most tiles contain 2–4 images from different dates: 53.21 % of tiles have a maximum date span of ≤ 5 days, and 91.07 % have a maximum span of ≤ 10 days. In iceberg-dense regions (65–80°S), 56.47 % of tiles span ≤ 5 days and 92.35 % span ≤ 10 days; in less dense regions (55–65°S), 89.09 % of tiles span ≤ 10 days. We have added the relevant explanation in the revised manuscript in L99-101. Referring to Koo et al. (2023), who reported that most icebergs in the Amundsen Sea sampling area drift at < 0.2 km/day (Figure S6), the short inter-image intervals and limited drift speed yield mosaics with good boundary and texture continuity. At this rate, the cumulative 10-day displacement is <2 km—negligible relative to the tile dimensions—so bergs are unlikely to exit a tile, and repeated detection of the same iceberg is unlikely.

We also considered the rare cases of fast-moving small icebergs being detected on adjacent dates, and we explain in the revised manuscript (L198–199) that these were removed by manual correction. For the two typical repeat cases shown in Figure S7, because icebergs tend to drift together in a relatively stable spatial arrangement under the combined influence of wind and currents, we retained only the set of icebergs with the most complete outlines (e.g., those in the red box of Sample Area 1

and the yellow box of Sample Area 2). As for the very few small icebergs counted twice due to rapid drift, their impact on the total number and area estimates for all Antarctic icebergs is negligible and can be ignored. In this way, the final iceberg count should truthfully and reliably reflect the actual distribution of icebergs in the study area.

[Figure]

Figure S5. Panel (a) distribution of the number of Sentinel-1 images per tile. Panels (b–d) histograms of the time span between acquisition dates for tiles in different latitude bands (55°S–80°S, 55°S–65°S and 65°S–80°S).

[Figure]

Figure S6. Drift speed distribution (biweekly average): most icebergs drift at speeds below 0.2 km/day, indicating relatively slow short-term movement (Koo et al., 2023).

[Figure]

Figure S7. Examples of fast-moving icebergs appearing twice in the mosaic imagery of the same tile.

L347: We -> we

We thank the referee for their careful correction and we have amended this in the revised manuscript.

L355-356: "in the West Antarctic region and in the East Antarctic region" -> It would be better to only specify Thwaites and Doston ice shelves and Holmes and Mertz ice shelves, without mentioning too ambiguous "West and East Antarctic regions".

We thank the referee for their valuable suggestions. In the revised manuscript, we have replaced "West Antarctic region" and "East Antarctic region" with the specific ice shelves Thwaites, Dotson, Holmes, and Mertz (L359).

L379-382: "In the Ross Sea sector, the iceberg proportion remained stable at around 16 % in 2018 and 2019, … remained relatively stable at approximately 20% over the six-year period." In those sentences, the "iceberg proportion" may indicate "the number of icebergs in each sector / the number of total icebergs in the Southern Ocean." However, I feel like this term "iceberg proportion" can be confused with "how much area (in percentage) is covered by icebergs (i.e., iceberg area / total ocean area of each sector)." Please consider rephrasing these sentences to clarify the meaning of the iceberg proportion. It could be good to discuss just the numbers (in Figure 11a), rather than the proportions (in Figure 11b).

We thank the referee for their correction. The term "iceberg proportion" in the text refers to the share of each region's iceberg count relative to the total number of icebergs in the Southern Ocean. To avoid ambiguity, we have clarified this definition (L383) and revised the wording accordingly in the revised manuscript.

L387: This is similar to the previous comment; please clarify the meaning of "total area." I believe this means the total area of icebergs.

We thank the referee for their correction. The term "total area" here refers to the cumulative iceberg area, and we have clarified this definition in the

revised manuscript (L391).

L394-401: I'm not sure that this part really "validates" the small iceberg formation mechanism. The authors just present the distance from large icebergs, and it does not provide any direct clues for the small iceberg formation mechanisms. I don't think this part is necessary.

We thank the referee for their valuable suggestions. We acknowledge that our study does not directly "validate" the formation mechanisms of small icebergs; therefore, in the revised manuscript we have modified the statement to: "In analyzing the distances between small and large icebergs, we further arrived at conclusions consistent with the formation mechanisms of small icebergs proposed by Tournadre et al. (2016)." The spatial distribution pattern of distances between small and large icebergs obtained in this study closely matches the findings of Tournadre et al. (2016), and we have additionally included the distribution near the Antarctic coastline to provide more comprehensive support for the formation mechanisms of small icebergs.

[Figure]

Figure S8. Average distance from icebergs in each grid to the nearest large iceberg. left panel: results from Tournadre et al. (2016); right panel:

results from our research.

**References**

Stern, A. A., Adcroft, A., and Sergienko, O.: The effects of Antarctic iceberg calving‐size distribution in a global climate model, JGR Oceans, 121, 5773‐5788, https://doi.org/10.1002/2016JC011835, 2016.

Koo, Y., Xie, H., Mahmoud, H., Iqrah, J. M., and Ackley, S. F.: Automated detection and tracking of medium-large icebergs from Sentinel-1 imagery using Google Earth Engine, Remote Sensing of Environment, 296, 113731, https://doi.org/10.1016/j.rse.2023.113731, 2023.

Tournadre, J., Bouhier, N., Girard‐Ardhuin, F., and Rémy, F.: Antarctic icebergs distributions 1992‐2014, JGR Oceans, 121, 327‐349, https://doi.org/10.1002/2015JC011178, 2016.

---

## Author Comment (AC2)

**Referee #2: Braakmann-Folgmann, Anne**

The research article "A Six-year circum-Antarctic icebergs dataset (2018-2023)" presents a novel and valuable dataset of iceberg population, distribution and area estimates for October in six consecutive years covering the whole Southern Ocean south of 55 deg (wherever Sentinel 1 EW data is available). It is the first study to include icebergs of all sizes with a minimum of 0.04 km2 and covering both open water and sea ice. Therefore, I consider this study novel, innovative and valuable for many downstream applications and future studies and recommend publication after some minor revisions listed below:

Thank you for your recognition of our study, and we are also very grateful for your detailed comments and valuable suggestions. Below are our point-by-point responses to each of your comments:

General: On zenodo, where the data is published, there is one section specifically for iceberg detection code and the iceberg sample set, but not for the iceberg vector outlines, which are the main dataset. I would suggest adding a paragraph on them explaining what the data contains and what units each variable comes in! Ideally, the units should also be added to the header within the dataset (e.g. area [km^2] rather than just area) or there should be a readme file with the same information added to the iceberg vector outlines zip file for ease of use.

We sincerely thank the reviewer for their valuable suggestion. We have noted that the current Zenodo page lacks detailed textual descriptions of the iceberg vector outlines files. In the updated Zenodo page (https://doi.org/10.5281/zenodo.16913262), we have added a relevant section providing explanations of the vector data, clearly defining the

physical meaning of each variable and its corresponding units, in order to enhance the readability and usability of the dataset.

L9/10: You don't mention mass as a geometric attribute, but that there is an uncertainty estimate for mass. As mass is not directly derived from the data, I would either leave it out or explain that mass is derived using a constant thickness and density.

To provide a comprehensive overview of the dataset in the abstract, we have adopted your second suggestion in the revised manuscript and now explicitly state that the mass is derived from the Volume/Area scaling parameterization of Iceberg Classes Model in Stern et al. (2016) under fixed-density assumptions (L10/11).

L12: The statement that this is related to A68 is not clearly backed up by your analysis or discussed in the paper. Either leave out or add more discussion

We thank the referee for the correction. We recognize that mentioning specific icebergs (such as A68) in the discussion of the overall Antarctic iceberg number change appears abrupt and lacks sufficient analytical support. Therefore, we have removed this part from the revised manuscript.

Table 1: I would suggest adding the studies by Wesche and Dierking and Barbat to the table

We sincerely thank the referee for the valuable suggestion. Our original intention was to summarize long-term iceberg distribution databases or data products, while the studies by Wesche & Dierking (2015) and Barbat et al. (2016), focus more on technical advances in iceberg detection

methods. However, to make our review of existing iceberg databases more comprehensive and complete, we have incorporated these studies into Table 1 in the revised manuscript as per your suggestion. Once again, we appreciate your thorough review and constructive comments.

Figure 1: This is a nice plot and clearly motivates why you picked October. However, did you just pick one location (indicated by the coordinates in the legend) for each surface type? And did e.g. the iceberg not move? From the text it is not clear at all whether this analysis was based on 1 pixel, 1 area or how many samples (area and number of images, locations) were used. Please explain.

We thank the referee for the detailed question. Yes, for the analysis in Figure 1, we selected a representative location for each surface type and indicated its specific coordinates in the legend. The iceberg sample was chosen as one that remained grounded without significant drift during the study period to ensure consistency in the time series analysis. This analysis was conducted based on a single pixel with precise latitude and longitude positioning, and we have added relevant explanations in the revised manuscript (L73–78) to improve clarity and reproducibility.

Figure 2: How does the iceberg classification result impact your iceberg thickness calculation? Isn't it solely based on altiberg? And the area/perimeter is independent of thickness?! So, I would suggest two parallel processing chains and merging them only for the mass (if I understand correctly).

Thank you for your suggestion. Initially, to simplify the workflow, we directly merged the two datasets and processed them either separately or jointly as needed in subsequent steps. In fact, the iceberg classification results do not affect thickness calculations, and the extraction of iceberg

area and perimeter does not rely on the Altiberg data. Therefore, our original approach may have caused some misunderstanding. In the revised version, we have updated the method for calculating iceberg thickness (L206–213), with thickness derived from iceberg area using the Volume/Area scaling parameterization of Iceberg Classes Model in Stern et al. (2016). Accordingly, Figure 2 has been modified to reflect this change.

L93: I assume most places are covered by several Sentinel 1 scenes within 1 month. How do you select which scenes to use and how do you ensure that icebergs are not missed or counted twice when they drift between scenes that are up to 30 days apart?

We sincerely thank the referee for this pertinent question. In our workflow, all Sentinel-1 HH-polarized scenes acquired within each 5° × 5° tile during the month were arranged chronologically and mosaicked in sequence, with later scenes replacing valid pixels from earlier ones. This procedure ensures that the composite image for each tile represents the most up-to-date spatial coverage while minimizing temporal gaps.

In practice, the temporal separation between images used for a given tile is short. For example, in 2021 over 91 % of all valid Antarctic tiles had a maximum acquisition-date span of no more than 10 days, and more than half had spans of 5 days or less in iceberg-dense regions (65–80° S) (Fig S1). Considering the low drift rates of most icebergs (< 0.2 km day$^{-1}$; Koo et al., 2023) (Fig S2), such short intervals mean that the likelihood of a single iceberg being detected twice within the same month is very small.

In addition, we accounted for the uncommon situation where rapidly drifting small icebergs might appear in images from adjacent acquisition dates. As clarified in the revised manuscript (L198/199), such duplicate

detections were identified and removed through manual inspection. For instance, in the two representative cases shown in Fig S3, icebergs were observed drifting in relatively stable clusters under the combined influence of wind and currents; in these cases, we retained only the set with the most complete outlines (e.g., those highlighted in the red box of Sample Area 1 and the yellow box of Sample Area 2). The few small icebergs that were inadvertently counted twice due to fast drift have an insignificant effect on the overall Antarctic iceberg count and area estimates, and thus can be disregarded.

[Figure]

Figure S1. Panel (a) distribution of the number of Sentinel-1 images per tile. Panels (b–d) histograms of the time span between acquisition dates for tiles in different latitude bands (55°S–80°S, 55°S–65°S and 65°S–80°S).

[Figure]

Figure S2. Drift speed distribution (biweekly average): most icebergs drift at speeds below 0.2 km/day, indicating relatively slow short-term movement (Koo et al., 2023).

[Figure]

Figure S3. Examples of fast-moving icebergs appearing twice in the mosaic imagery of the same tile.

I am missing an explanation somewhere in your methods how you define each iceberg object. My understanding is that you classify each superpixel into iceberg or not and then do a manual correction. When do you merge neighbouring superpixels that were classified as iceberg into

one iceberg? And have you tested how far apart two icebergs need to be to separate them? Or does each superpixel need manual redrawing of the outline anyway before it becomes an iceberg?

Thank you for raising this valuable question. In our methodology, the classification results—i.e., the superpixels identified as icebergs—are first converted into a binary mask. After hole-filling and denoising, we directly apply connected-component labeling to the binary image. This step automatically merges all adjacent iceberg-classified superpixels into a single iceberg object. Two icebergs are recognized as separate objects only if there is at least one non-iceberg superpixel between them, and no additional distance threshold is applied for segmentation. On this basis, we perform only necessary manual corrections to the merged iceberg boundaries to address false detections, omissions, or boundary deviations, rather than redrawing the outline of each superpixel individually. Therefore, the aggregation or separation of iceberg objects is entirely determined by pixel-level connectivity. We have added this clarification in the revised manuscript (L189–192) to enhance the transparency and reproducibility of the methodology.

L145: What do you mean by sample points here? Are these the superpixels derived by SLIC? Or individual pixels? Or merged icebergs?

We thank the referee for the question. Here, the term "sample points" refers to the superpixels extracted from Sentinel-1 imagery using the SLIC algorithm, and we have added a corresponding clarification in the revised manuscript (L153/154).

L178/179: How do you identify which icebergs are counted twice? Most have rather generic shapes or can rotate and break up in between.

We thank the referee for this question. As noted above, because the

mosaics are constructed from Sentinel-1 scenes arranged in chronological order and Antarctic icebergs generally drift slowly over short time intervals, the likelihood of the same iceberg appearing in two adjacent scenes and being double-counted is very low (Fig S1-3). The potential counting errors caused by such cases are far smaller than those arising from misclassification or omission during the detection process.

In our workflow, we therefore focus on identifying clusters of icebergs that move together as a group under the combined influence of winds and currents. For these clusters, we manually remove repeated instances of the same iceberg, prioritizing the version with the most complete and accurate outline. In contrast, very small icebergs with atypical texture or shape features—particularly those that drift rapidly—are not targeted for individual de-duplication, as their contribution to total counts and areas is negligible.

L182: As you use a constant thickness and density for all icebergs, I think it would be better to just assign those parameters to individual icebergs that are actually derived from the data (i.e. area, perimeter, axes, coordinates) and only use the thickness and density to calculate the overall mass of icebergs in each year. For this application your assumptions seem fair and some of the uncertainty will average out, whereas the smaller bergs will certainly be thinner than you assume and some giant bergs will be thicker, so assigning the average thickness to each berg seems like an unnecessary stretch.

Thank you for your valuable suggestion! We acknowledge that assuming a constant thickness is meaningful for estimating the overall Antarctic iceberg mass but often introduces large biases at the individual iceberg scale. In the revised version, we applied the Iceberg Classes Model to estimate both the mass of individual icebergs and the annual total iceberg

mass across the circumpolar region (Gladstone et al.,2001; Stern et al., 2016). Following the parameterization scheme of Nong et al. (2025), the model provides an area-volume power-law relationship, with iceberg thickness constrained to a maximum of 250 m. This constraint implies that an iceberg with a thickness of 250 m corresponds to an area of 0.67 km2. For icebergs smaller than this threshold, volume is calculated directly from the power-law relationship ($V_{iceberg}=7.64A^{1.26}$), whereas for larger icebergs, volume is derived by multiplying the area by the fixed thickness of 250 m. Assuming an average density of 850 kg/m3(Silva et al., 2006), the mass of each iceberg and the circumpolar total are then obtained accordingly.

In addition, we compared the impacts of two methods on estimating the total Antarctic iceberg mass: the fixed-thickness method (232 m) and the segmented method (using the power-law relationship for icebergs with an area < 0.67 km², and a fixed thickness of 250 m for those with an area ≥ 0.67 km²). The results show that (Figure S4), compared with the fixed-thickness method, the segmented method yields smaller masses for small icebergs (area < 0.67 km²) but larger masses for large icebergs (area ≥ 0.67 km²), indicating that the new approach avoids pulling the two ends toward the middle and instead enlarges the mass contrast between small and large icebergs. Moreover, the total Antarctic iceberg mass estimated by the segmented method is greater than that from the fixed-thickness method, and the variation in large-iceberg mass closely follows that of the total mass, further demonstrating that large icebergs dominate the overall Antarctic iceberg mass.

[Figure]

Figure S4. Comparison of Antarctic iceberg mass estimation methods: fixed-thickness method and segmented method, with shaded area representing the uncertainty range of the segmented method.

L301-303: I am surprised that BYU/NIC miss so many icebergs. Are most of the ones missing from their database around the threshold of 5 km? Or are you sure you weren't counting some double? Or accidentally merged two smaller bergs into one bigger one?

We thank the referee for this question. To address the possibility of double counting or incorrect merging, we manually cross-checked all icebergs in our database with a major axis greater than 5 km against the original Sentinel-1 mosaicked images for October 2021. This verification confirmed that no duplicate counts or erroneous mergers occurred.

Furthermore, our one-by-one comparison with the BYU/NIC Statistical Database (v7.1) shows that the large number of icebergs absent from BYU/NIC is not concentrated around the 5 km threshold (Fig S5-6). Instead, these undetected icebergs span a range of sizes, with most having major axes of 5-9 km and areas of 0-20 km². Many of them are located in

front of ice shelves where dense sea ice cover can obscure detection by passive microwave or scatterometer sensors. As discussed by Budge and Long (2018), such coarse-resolution sensors and the challenging coastal environment can cause even large icebergs to be missed. Therefore, the discrepancies between the two datasets primarily reflect limitations in BYU/NIC's detection capability rather than over-counting or merging errors                                                                                                      in ours.

[Figure]

Figure S5. Histogram distribution of the area and major axis length of icebergs (>5 km) missed by the BYU/NIC dataset.

[Figure]

 Icebergs(> 5 km) missing from BYU/NIC database
 Icebergs detected by BYU/NIC database

Figure S6. Examples of icebergs (>5 km) detected (blue) and missed (red) by the BYU/NIC dataset, with Sentinel-1 mosaics as background.

L332 Small icebergs are more influenced by wind, not by currents.

We sincerely thank the referee for the correction. As small icebergs are relatively small in size, their movement is more influenced by wind than by ocean currents. We have revised the manuscript accordingly, replacing "coastal currents" with "wind and coastal currents" to more accurately describe the drift mechanism of small icebergs.

L350-352: It does not make sense to analyse trends in mass if your

thickness and densities are constant. You can analyse trends in area, but the mass is just a multiple of your area, so just leave this section out.

Thank you for your comment. We acknowledge that analyzing mass trends lacks practical significance when both iceberg thickness and density are held constant. However, in the revised manuscript, we have updated the method for calculating mass and no longer use a fixed thickness. Therefore, we have retained the analysis of mass trends in the revised version.

Figure 8: Very nice figure!

We thank the referee for the positive feedback on Fig. 8.

L417 Thickness and density will also depend on the calving location/mother ice shelf (Dowdeswell and Bamber, 2007 and Ligtenberg et al. 2011).

We thank the referee for the correction. We have added this point in the revised manuscript (L421/422).

Figure 10: Add a comment that the y axis in c starts at 80 % - it's easy to miss

We thank the referee for the suggestion. We have added the caption for Fig 10 in the revised manuscript.

**References**

Koo, Y., Xie, H., Mahmoud, H., Iqrah, J. M., and Ackley, S. F.: Automated detection and tracking of medium-large icebergs from Sentinel-1 imagery using Google Earth Engine, Remote Sensing of Environment, 296, 113731, https://doi.org/10.1016/j.rse.2023.113731,

2023.

Gladstone, R. M., Bigg, G. R., and Nicholls, K. W.: Iceberg trajectory modeling and meltwater injection in the Southern Ocean, J. Geophys. Res., 106, 19903–19915, https://doi.org/10.1029/2000JC000347, 2001.

Stern, A. A., Adcroft, A., and Sergienko, O.: The effects of Antarctic iceberg calving-size distribution in a global climate model, JGR Oceans, 121, 5773–5788, https://doi.org/10.1002/2016JC011835, 2016.

Nong, M., Liu, X., Li, T., Zhang, B., Liang, Q., Zheng, L., Zhao, T., and Cheng, X.: Characterizing Nearshore Icebergs in front of th e Dalk Glacier, East Antarctica by UAV Observation, https://doi.org/ 10.5194/egusphere-2025-1884, 10 June 2025.

---

## Author Response (AR2)

**Response:**

Dear Editor,

We sincerely thank the editor and reviewers for their thorough evaluation and constructive comments. Their feedback is highly valuable for improving the quality and clarity of our manuscript and dataset. We have carefully considered all suggestions, and in the revised version we provide detailed clarifications, additional quantification of uncertainties, and expanded discussions to better address the concerns raised. For further details, please refer to our point-by-point responses to your comments.

Best,
Teng

In particular, both reviewers ask for clarification on the iceberg detection/segregation, on the way thickness is estimated and used, and how double detections are avoided/iceberg movement is handled. Note that a careful description and discussion of errors and limitations of your dataset is essential for it to be useful for the community.

The reviewers also request further details on the comparison with the BYU/NIC dataset. A discussion of differences in the detection abilities of the approaches behind the validation datasets will be useful for the readers. The suggestions of referee #2 for the zenodo dataset metadata/description greatly improves user-friendliness of the data.

We sincerely thank you and the two reviewers for your valuable comments, which have greatly improved the quality of our manuscript.

In the revised manuscript, we have made the following improvements: (i) conducted a detailed quantification and discussion of the uncertainties caused by duplicate iceberg detections (L244-253 and L292-297); (ii) added further discussion of the comparison with the BYU/NIC dataset and analyzed differences in the detection abilities of the respective approaches (L322-331); (iii) revised the description of the ensemble incremental random forest parameter settings to make the procedure clearer and easier to follow (L153-172); and (iv) included six supplementary figures (Figs. S1-S6), which are now explicitly referred to in the manuscript text.

Lastly, is this a manually made track changes document? E.g., lines 165ff (not crossed out) suggest so, and in many places, entire paragraphs/sentences are crossed out and added back below with seemingly only slight text changes. It is a lot of work for the reviewers and the editorial team to find and identify the actual text changes made in reply to their comments. The review process is based on voluntary work entirely, and spending this extra effort is too much to ask from the reviewers and editorial team. Therefore, we expect a track changes document that allows to track text changes in a targeted way.

When resubmitting, please provide a track changes document where only changed text is highlighted, using e.g. latexdiff or the track changes/record changes option available in numerous office suite software. A resubmission with some other form of track change documents will not be accepted.

In this resubmission, the track changes document was generated automatically using latexdiff, and we sincerely apologize for the earlier submission of an improper track changes document. We truly appreciate

your patience and guidance, and we are committed to following the proper procedures in all subsequent submissions.

Additional editor comments:

- In your reply to the reviewers, you use descriptions such as likelihoods/uncertainties are "very small", or "very low", with "insignificant effect", these are vague terms. Can you quantify these uncertainties/likelihoods?

This being a data documentation publication, information about uncertainties in your dataset and comparisons to other datasets are crucial for the reader and data user. Any uncertainties (also as a result of method choices/limitations) need to be stated, if possible quantified, and discussed in the validation and uncertainty or discussion section. This needs to be improved before resubmitting.

We sincerely thank the editor for highlighting this important point. Below, after adding new experiments, we provide a revised response to the reviewers' question. The specific questions are as follows:

*Anonymous Referee #1:L339-349: I wonder if the total number of icebergs here and in Table 5 is the "true" number of icebergs. That is, if an iceberg is detected in two different Sentinel-1 scenes, how is this iceberg counted? This iceberg could be counted in duplicate, as the methods proposed in this study can only "detect" icebergs but cannot "track" identical icebergs. This could not be so significant because the authors used mosaiced data, but there is a possibility that the same icebergs are detected in duplicate (or some icebergs are missed) due to their drift even over a short period. It would be worthwhile to mention this issue and include any relevant discussion about it.*

*Referee #2: Braakmann-Folgmann, Anne: L93: I assume most places are covered by several Sentinel 1 scenes within 1 month. How do you select which scenes to use and how do you ensure that icebergs are not missed or counted twice when they drift between scenes that are up to 30 days apart?*

Response: First, regarding image selection, we did not manually choose or filter scenes. During the image acquisition stage, all Sentinel-1 HH-polarized images within each tile were sorted in ascending order of acquisition time and mosaicked sequentially in Google Earth Engine using the mosaic function. Later-acquired images overwrite valid pixels from earlier ones, thereby filling gaps at the beginning of the month and producing a spatially continuous composite that represents the most recent observations. Statistics show that most tiles contain 2-4 images from different dates: in each year, more than 50% of tiles have a maximum time span of less than 5 days, and more than 90% have a maximum span of less than 10 days (Fig. R1).

We fully agree with the reviewer's concern that our method cannot track individual icebergs. Even after manual corrections, only large icebergs with distinct shape or texture features can be reliably checked, while smaller ones may still be subject to duplicate counting. To quantify this effect, we used the 2021 Antarctic mosaic on the Google Earth Engine platform and extracted acquisition dates (YYYYMMDD) for each pixel (Fig. R2). For every iceberg smaller than 10 km² (we consider larger ones to be fully resolved through manual correction), we assigned the centroid pixel's acquisition date and computed the distance to the nearest pixel acquired later in time. If this distance was smaller than the product of the date difference and the mean drift speed, the iceberg was flagged as a potential duplicate. Previous regional studies report mean drift speeds of

about 3-7 km d⁻¹ (Hamley and Budd, 1986; Collares et al., 2018; Barbat et al., 2021; Orheim et al., 2023), and we adopted 5 km d⁻¹ as a representative value. In 2021, a total of 1,757 icebergs were identified as potential duplicates, representing 3.36% of the total count and 655 km² (1.25%) of the total area. We therefore assign 2% as the uncertainty contribution from duplicate counting, representing a conservative cross-year upper limit. This quantification procedure has been added to the revised manuscript (L245-253).

[Figure]

**Figure R1.** Panel (a) distribution of the number of Sentinel-1 images per tile. Panels (b-d) histograms of the time span between acquisition dates for tiles in different latitude bands (55°S-80°S, 55°S-65°S and 65°S-80°S).

[Figure]

**Figure R2**. Spatial distribution of detected icebergs and potential duplicates around Antarctica in October 2021. Colors indicate the acquisition dates of Sentinel-1 EW scenes used to construct the monthly mosaic, while yellow and red points mark iceberg detections and potential duplicate icebergs, respectively.

- In the replying (some of) these to a manuscript supplement? If so, please refer to these in the manuscript text.

We thank the editor for this reminder. In the revised submission, we have added the relevant figures as supplementary material (a total of six figures) and have explicitly referred to them in the manuscript text where

appropriate (e.g., Figs. S1-S6 in the Supplement).

Minor comments:

- Construction of Incremental random forest classifiers (added after a request from reviewer 1): Please pay extra attention to this section, and ensure the process is transparent and understandable for the reader.

We thank the editor for this valuable suggestion. In the revised manuscript, we have simplified and clarified the description of the construction and parameterization of the incremental random forest classifiers. The revised text avoids redundant phrasing and provides a clearer step-by-step explanation of the procedures for classifier weighting and threshold setting.

- Some comments seem not considered, e.g. reviewer 2 for Fig. 10.

We thank the editor for the reminder. In the revised manuscript, we have added the note "Note that the y-axis in (c) is truncated at 80% for clarity" to the caption of Fig. 10.

**References**

Hamley, T. C. and Budd, W. F.: Antarctic Iceberg Distribution and Dissolution, J. Glaciol., 32, 242–251, https://doi.org/10.3189/S0022143000015574, 1986.

Collares, L. L., Mata, M. M., Kerr, R., Arigony-Neto, J., and Barbat, M. M.: Iceberg drift and ocean circulation in the northwestern Weddell Sea, Antarctica, Deep Sea Research Part II: Topical Studies in Oceanography, 149, 10–24, https://doi.org/10.1016/j.dsr2.2018.02.014, 2018.

Barbat, M. M., Rackow, T., Wesche, C., Hellmer, H. H., and Mata, M. M.: Automated iceberg tracking with a machine learning approach applied to SAR imagery: A Weddell sea case study, ISPRS Journal of Photogrammetry and Remote Sensing, 172, 189–206, https://doi.org/10.1016/j.isprsjprs.2020.12.006, 2021.

Orheim, O., Giles, A. B., Moholdt, G., (Jo) Jacka, T. H., and Bjørdal, A.: Antarctic iceberg distribution revealed through three decades of systematic ship-based observations in the SCAR International Iceberg Database, J. Glaciol., 69, 551–565, https://doi.org/10.1017/jog.2022.84, 2023.

---

## Author Response (AR3)

Response:

Dear Editor and Reviewers,

We would like to thank you for your time and effort in reviewing our revised manuscript. We are grateful for the positive feedback and the constructive comments provided in this second round of review. We have carefully considered all the suggestions and have made corresponding changes to the manuscript. Below is a point-by-point response to the reviewers' comments. All line numbers refer to the tracked changes document.

Best,
Teng

Response to Reviewer #1

- Figure 2: In the image segmentation part, should "SNIC" be replaced with "SLIC"?

We thank the reviewer for this correction. We confirm that we utilized the SLIC (Simple Linear Iterative Clustering) algorithm, and the typo in Figure 2 has been corrected in the revised manuscript.

- L169: It would be good to refer to Figure S2 here.

We thank the reviewer for the suggestion.We have refer to Figure S2 in the corresponding sentence (L162).

- L174-179: Can the authors add any supplementary figures in determining the decision thresholds? Figure S2 was super helpful for understanding how the random forest settings are determined based on the out-of-bag error analysis, so it would also be good to show examples of P-R and ROC curves in the supplement to help readers' understanding.

We thank the reviewer for this constructive suggestion. We have added a new supplementary figure, Figure S3, which presents examples of the ROC and P-R curves, and also added a reference to Figure S3 in the main text (L168).

- L427: "a lower minimum detection threshold" -> It would be good to specify "a lower threshold for minimum iceberg size."

We have revised the phrase to "a lower threshold for minimum iceberg size" in the manuscript as suggested (L370).

Response to Reviewer #2

- Figure 1: Please also mention in the caption of Figure 1 that the time series refer to 1 pixel each.

We appreciate the reviewer's suggestion to clarify the spatial scale of Figure 1. We have revised the caption to explicitly state: Each time series corresponds to a single pixel.

- Figure 2: What is meant by Iceberg Ocean model? Is this what you call iceberg classes model in the abstract? Please use a consistent name. I would actually just call it a scaling.

We apologize for the inconsistency and confirm that 'Iceberg Ocean model' and 'Iceberg Classes Model' refer to the same concept. We agree that 'Iceberg Size Scaling' is a more accurate term and have standardized its usage throughout the manuscript, including Figure 2 and the main text (L10 and L207).

-L182-183: If you optimise the parameters for each year, the method is not tested on its generalisation capabilities! However, L318 says you leave one year out for testing in a cross-validation? Please clarify how exactly you do it and how much the parameters are tuned to each year.

We thank the reviewer for the constructive comments regarding our validation strategy. We acknowledge that, because our original approach optimized model parameters separately for each year, the previous description of the "leave-one-out cross-validation" procedure was ambiguous and potentially illogical.

To clarify the exact procedure: each year we generate approximately 2,000 manually labeled superpixel samples, of which 1,200 are used for training, 400 for parameter tuning (validation set), and another 400 as an test set. Taking 2018 as an example, after completing parameter optimization for that year, we previously attempted to use the test samples from other years as training data and then evaluated the model using the 2018 test set. Although this result suggested that the model parameters and accuracy did not vary substantially among years—indicating a certain degree of stability—we recognize that because the parameters were optimized

specifically for one year, applying them directly to other years' data cannot rigorously demonstrate the model's generalization ability.

Therefore, in the revised manuscript (L280-290), we have replaced the previous approach with a strict year-by-year accuracy assessment. Under this revised strategy, the model optimized for a specific year (e.g., 2018) is evaluated exclusively using the independent test set from that same year, ensuring full independence and methodological consistency.

In addition, to illustrate the magnitude of parameter adjustments across years, we provide the optimal parameters for each year (including classifier weights and decision thresholds) in the newly added Table S1 in the Supplement. As shown in Table S1, these parameters exhibit only minor interannual variations, indirectly demonstrating the stability of the model. In future work, we plan to integrate multi-year datasets to develop a unified, year-independent model capable of processing new data without retraining.

-L 221: Nong et al (2025) is not peer reviewed yet. If you base your methods on this paper, please describe and verify it yourself.

We thank the reviewer for this valuable comment. In the revised manuscript, we have removed the citation to Nong et al. (2025) and have provided a clear description of how we derive the area-volume relationship for small icebergs directly from the 10-class iceberg classification scheme proposed by Stern et al. (2016) (L210–216).